# Efficacy Evaluation of Oregano Essential Oil Mixed with *Bacillus thuringiensis israelensis* and Diflubenzuron against *Culex pipiens* and *Aedes albopictus* in Road Drains of Italy

**DOI:** 10.3390/insects13110977

**Published:** 2022-10-25

**Authors:** Athanasios Giatropoulos, Romeo Bellini, Dionysios T. Pavlopoulos, George Balatsos, Vasileios Karras, Fotis Mourafetis, Dimitrios P. Papachristos, Filitsa Karamaouna, Marco Carrieri, Rodolfo Veronesi, Serkos A. Haroutounian, Antonios Michaelakis

**Affiliations:** 1Benaki Phytopathological Institute, 8 Stefanou Delta Street, Kifissia, 14561 Athens, Greece; 2Centro Agricoltura Ambiente “G. Nicoli”, Via Sant’Agata 835, 40014 Crevalcore, Italy; 3Faculty of Animal Sciences, Agricultural University of Athens, Iera Odos 75, 11855 Athens, Greece

**Keywords:** mosquito larval control, insecticides mixtures, essential oil, oregano oil, carvacrol, diflubenzuron, *Bacillus thuringiensis israelensis*

## Abstract

**Simple Summary:**

The continuous use of synthetic insecticides against mosquitoes has led to resistance problems, while the use of bio-insecticides may face limitations in their actual use in the field. Alternatively, binary mixtures of botanicals with conventional synthetic or bio-insecticides have been proven, primarily on the laboratory scale, to be potential eco-friendly mosquito larvicides. In the current study, we investigated in the field the joint action of a carvacrol rich oregano Essential Oil (EO) with two conventional insecticides, namely, the insect growth regulator diflubenzuron and the bio-insecticide *Bacillus thuringiensis israelensis* (*B.t.i.*), in road drains of Crevalcore city, Italy, against *Culex pipiens* and *Aedes albopictus.* The results showed that the application of mixtures of EO with diflubenzuron or *B.t.i.* provided sufficient control of immature mosquito population for a 2–3 week period. These findings suggest the potential of mixing carvacrol-rich EO with diflubenzuron and *B.t.i.* as an efficient eco-friendly alternative to single insecticide applications in road drains against *Cx. pipiens* and *Ae. albopictus* larvae.

**Abstract:**

Mosquito management programs in the urban environment of Italian cities mainly rely on larval control with conventional insecticides, primarily targeting the road drains that constitute the principal mosquito breeding sites encountered in public. The repeated utilization of synthetic insecticides may have adverse effects on non-targets and lead to resistance development issues, while the performance of biopesticides encounters limitations in field use. Botanical insecticides as single larval control agents or in binary mixtures with conventional insecticides have been extensively studied in the laboratory as an effective and eco-friendly alternative mosquito control method with promising results. The study herein concerns the investigation, for the first time under realistic conditions in the field, of the joint action of a carvacrol-rich oregano Essential Oil (EO) with two conventional insecticides, namely, the insect growth regulator diflubenzuron and the bio-insecticide *Bacillus thuringiensis israelensis* (*B.t.i.*), in road drains in Crevalcore city, Italy, against *Culex pipiens* and *Aedes albopictus*. According to the obtained results, the application of both plain EO and its mixtures with diflubenzuron and *B.t.i.* exerted very high efficacy in terms of immature mosquito population reduction over a two-week period. Three weeks after treatment, the performance of the oil and its mixtures diminished but remained high, while the addition of diflubenzuron potentiated the persistent action of the oil against *Cx. pipiens*. These findings are indicative of the potential of mixing carvacrol-rich EO with diflubenzuron and *B.t.i.* as an efficient eco-friendly alternative to mono-insecticide applications in road drains against *Cx. pipiens* and *Ae. albopictus* larvae.

## 1. Introduction

Mosquitoes constitute the most important group of medically important insects, and represent a considerable threat to human and animal health due to their ability to transmit a broad variety of viruses and other pathogens [1]. *Aedes albopictus* (Skuse 1894), widely known as the Asian tiger mosquito, is the most invasive mosquito species in the world and a field vector of dirofilarial heartworms and numerous viruses such as Chikungunya, Dengue, and Zika [2]. Following its first detection in Albania in 1979 [3], *Ae. albopictus* has been introduced and established in many European countries, mainly those of the Mediterranean basin and eastern Black Sea [4]. After its first introduction into Italy in 1990, today *Ae. albopictus* is widespread in every region of the country [5], and was conceivably involved in the large Chikungunya outbreaks of 2007 and 2017 [6] as well as the first autochthonous dengue outbreak of Italy in 2020 [7]. In addition, the common house mosquito *Culex pipiens* (Linnaeus 1758) is a complex species native to Europe and the principal vector of West Nile virus (WNV) infection outbreaks [8,9]. Europe experienced its first large-scale epidemic of WNV in 1996 in Romania; subsequently, several new cases of the disease in humans have been reported from southern and southeastern European countries [10]. In Italy, *Cx. pipiens* is ubiquitously distributed throughout the country [11] and has caused several seasonal WNV outbreaks, which have been recorded almost annually since 2008 [12]. 

Among the different mosquito control strategies, larval control by targeting the aquatic immatures constrained in water bodies has proven the most effective, proactive, target-specific, and safe approach [13,14]. The role of mosquito larval control for the prevention and control of WNV infection outbreaks is considered as part of an Integrated Vector Management (IVM) program to keep adult mosquito populations at density levels below those posing a public health risk [8]. In this respect, larvicidal applications in catch basins or road drains have been proposed as standard control measures in public areas in the context of practical management plans against *Ae. albopictus* in Europe [6]. Road drains constitute highly suitable sites for mosquito larvae growth, as they contain water along with many urban human byproducts that greatly pollute the respective micro-environments [6]. According to Guzzetta et al. [15], routine larviciding of public catch basins in European cities with temperate climates can limit the risk of autochthonous transmission as well as the size of potential epidemics caused by *Ae. albopictus*. 

In Europe, larval control with conventional synthetic or bio-insecticides has long been used and is currently preferred as the primary mosquito control tool in mosquito management programs [16,17]. Currently, larviciding relies on a limited number of approved active substances for mosquito control in the European market under the EU Regulation 528/2012, i.e., the microbial insecticides *Bacillus thuringiensis* subsp. *israelensis* and *Bacillus sphaericus*, juvenile hormone mimics *s*-methoprene and pyriproxyfen, and chitin synthesis inhibitor diflubenzuron [18,19]. In Italy, larvicide treatment of road drains has resulted in the strong inhibition of adult emergence of *Ae. albopictus* and *Cx. pipiens* in diflubenzuron-treated catch basins [20]. Additionally, several diflubenzuron-based formulations have exerted high efficacy and good insecticide persistence against *Ae. albopictus* and *Cx. pipiens* in the Italian road drain system [21]. Finally, in the terms of an integrated pest control strategy against the tiger mosquito in Northern Italy, the utilization of *Bacillus thuringiensis israelensis* (*B.t.i*) and *Bacillus sphaericus* granules [22] for larvicidal treatments in both public and private catch basins has effectively reduced *Ae. albopictus* populations.

It must be noted, however, that continuous utilization of synthetic chemical insecticides may result in the development of resistance in mosquitoes [23,24]. In this respect, high levels of resistance of *Cx. pipiens* to diflubenzuron and focal distribution of diflubenzuron resistance mutations in *Cx. pipiens* have already been reported in Northern Italy [25,26]. On the other hand, it is well established that plant-derived products, e.g., essential oils (EOs), are less prone to inducing resistance compared to synthetic insecticides, as they contain a large number of active ingredients with multiple mechanisms of action on insects [27,28]. Thus, a constantly growing number of botanical EOs and plant extracts have been tested in various laboratories against mosquito larvae, showing their high efficacy through various modes of action [29,30]. Carvacrol and carvacrol-containing EOs derived from Lamiaceae plant species such as *Origanum* spp. have been reported to display significant larvicidal effects in laboratory bioassays against *Ae. albopictus* and *Cx. pipiens* [31,32,33].

Laboratory scale experiments have revealed several binary mixtures composed of botanical and synthetic insecticides or biopesticides that display promising activities as a result of synergistic or additive toxic effects against mosquito larvae [34,35,36,37,38]. The utilization of these binary systems comprises an eco-friendly strategy for the control of mosquito larvae that may increase overall efficacy, minimize the possibility of resistance development, and lead to lower risk for non-targets and humans by enabling dose reduction [36,39]. Although the larvicidal properties of EOs against mosquitoes have been widely explored in laboratory conditions [27,29], there is a lack of potency evaluation in field trials, which may be attributed to the light and heat instability of phytochemicals compared to synthetic insecticides [18,39]. 

In a two-week field trial implemented during the summer of 2018 in catch basins of the urban area of Crevalcore (Northern Italy), which constitute common mosquito breeding sites, both crude and emulsified carvacrol-rich oregano oils (*Origanum vulgare* ssp. *hirtum*) proved to be efficient larvicidal agents against *Cx. pipiens* and *Ae. albopictus* [40]. As a follow up, we report herein the results of a complementary field study performed in the summer of 2019 in the road drains of the same urban area of Crevalcore, Italy, with the aim of assessing the effectiveness of carvacrol-rich crude oregano EO as a mixture with diflubenzuron and *B.t.i.* against *Ae. albopictus* and *Cx. pipiens* populations. The study focused on the impact of diflubezuron and *B.t.i.* on the performance of oregano EO.

## 2. Materials and Methods

### 2.1. Chemicals Tested

EO of oregano (*Origanum vulgare* var. *hirtum*, Lamiaceae) originated from Kilkis prefecture of Greece. and was kindly donated by Ecopharm Hellas SA. According to its chemical analysis [40], carvacrol constituted the oil’s prevailing component (75.1%). Diflubenzuron liquid suspension concentrate (SC) formulation (DU-DIM 15 SC, 15% *w*/*v*) was provided by Arysta LifeScience. Liquid suspension concentrate (SC) formulation of *Bacillus thuringiensis* subsp. *israelensis* (*B.t.i*.) Serotype H14, Strain AM65-52 (Vectobac 12AS, 11.61% content of a.s.) was purchased from Sumitomo Chemical Agro Europe SAS.

### 2.2. Experimental Setup

The field trial was implemented in the urban area of Crevalcore, Italy, for three weeks, from the 4th to the 25th of July 2019. The application of crude carvacrol-rich oregano EO and the applied doses were determined on the basis of our previous results [40]. The first step of the trial involved the random selection of 29 road drains (dimensions 40 × 40 cm) colonized by mosquito larvae/pupae. The presence of at least one mature larva or pupa per water sample dipper was sufficient to consider the drain as colonized by mosquito larvae/pupae. The drains were closed hydraulic systems maintaining highly polluted water with suspended and floating materials. Then, seven drains were treated with 10 mL of oregano EO per drain, six with 10 mL oregano EO + 0.35 mL diflubenzuron liquid SC formulation (diluted in 30 mL of water) per drain (EO+DFB), and six with 10 mL oregano EO + 0.3 mL SC formulation of *B.t.i.* (diluted in 30 mL of water) per drain (EO+*B.t.i.*), with ten were left untreated as a control. The diflubenzuron-based formulation was applied at the recommended dose on the label for closed hydraulic systems. The applied dose of *B.t.i.* formulation was ten times higher than the average dose recommended on the label for open polluted water bodies due to the highly polluted water in closed hydraulic systems. The experimental road drains were sampled using a standard dipper of 0.5 lt capacity by collecting two water samples, waiting 2 min between the two samples. The collected mosquito larvae and pupae were transferred to the laboratory for the determination of age category, i.e., 1st–2nd instar larvae, 3rd–4th instar larvae and pupae, and their species. Mosquito samplings were performed just before the treatment (day 0, pre-treatment) and every week at 7, 14, and 21 days post-treatment. The water depth and temperature were measured for each drain and each sampling time point, along with the meteorological data (temperature, precipitation) during the trial period, which were recorded from a weather station located 1.7 Km from the study area (Sant’ Agata Bolognese weather station).

### 2.3. Data Analysis

The efficacy evaluation of the larvicidal treatments (EO, EO+DFB, EO+*B.t.i.*) in road drains against mosquitoes was determined through assessment of immature mosquito population reduction (%). This was calculated for each sampling day using Mulla’s formula [41], which allows the determination of natural changes in mosquito populations for both treated and untreated sites:% Population reduction = 100 − [(C1/T1) × (T2/C2)] × 100

In the above formula, C1 is the mean number of mosquitoes per drain at the control site pre-treatment; C2 is the mean number of mosquitoes per drain at the control site post-treatment, T1 is the number of mosquitoes in each drain at the treatment site pre-treatment, and T2 is the number of mosquitoes in each drain at the treatment site post-treatment. 

The mean (corrected) % of immature population reduction (+/− SEM) in each mosquito species and the total per drain were calculated. Data regarding the rate of population reduction at each sampling time point post treatment were analyzed with the non-parametric Kruskal–Wallis test (k samples) at α = 0.05. The SPSS 21.0 statistical package was used for data analysis.

## 3. Results and Discussion

During the field trials, the immature mosquito population in the experimental drains was dominated by larvae of all stages (L1–L4), whereas a small number of pupae were detected (Figure 1 and Figure 2); hence, the population composition was appropriate for the evaluation of larvicide treatments effects for the EO of oregano and its mixtures with diflubenzuron and *B.t.i.* targeting the larval mosquito stage. All experimental drains contained water throughout the field study, with a mean water depth of 17.6 cm (±0.6) and a mean water temperature inside the drains of 26.5 (±0.3) °C. Collected mosquito species belonged to *Cx. pipiens* and *Ae. albopictus,* as reported by Evergetis et al. [40] in mosquito samplings from catch basins treated with the same EO in the same area one year earlier in 2018. According to Figure 1 and Figure 2, *Cx. pipiens* was more abundant compared to *Ae. albopictus*. It must be noted, however, that both species were present in the untreated control of road drains at each sampling time point throughout the pre- and post- treatment trial using the larvicidal agents. 

From 4 July (treatment day 0) until 25 July when the trial was ended (after three weeks, on day 21), the mean air temperature was 20.3–26.8 °C; five rainfall events occurred, registering 2.5 to 36 mm rain (Figure 3). 

The application of the EO and its mixtures with both insecticides exerted very high efficacy against *Cx. pipiens* and *Ae. albopictus* for 7 and 14 days post-treatment, displaying a 97.4–100% population reduction (Table 1). The results reported herein reveal the two-week strong persistent action of larvicidal treatments in road drains with carvacrol-rich EO against immatures of *Cx. pipiens* and *Ae. albopictus,* as previously reported by Evergetis et al. [40]. The performance of EO and EO+*B.t.i.* against *Cx. pipiens,* as well as of all tested applications against *Ae. albopictus* in three weeks period was relatively lower although still high, providing a 72.4–83.1% reduction of mosquito population. On the contrary, the efficacy of the EO mixture with diflubenzuron against *Cx. pipiens* maintained very high (96.5%) population reduction for three weeks after treatment, indicating a higher persistence capacity over the entire period of testing compared to the application of plain EO. This result can be explained by the chitin synthesis inhibitory activity of diflubenzuron. 

Residual action comprises an important element for larvicidal treatments in control programs against mosquito larvae, as long-lasting treatments can potentially reduce the cost of larvicidal field applications, resistance development, and impacts on non-target organisms [17]. In this respect, efficacy in terms of the population reduction of the larvicidal applications with the EO, EO+*B.t.i.*, and EO+DFB did not differ significantly during the post-treatment weekly samplings for either mosquito species (Table 1). Nevertheless, non-significant statistical differences between applications observed in certain cases may be attributed to the high variability in mosquito populations between the experimental road drains (replicates); the latter constitutes a study limitation due to the fact that the trial was conducted under realistic environmental conditions in the field against wild mosquito populations. In this respect, the population reduction in each treatment plot (EO, EO+*B.t.i.*, EO+DFB) at each sampling day was corrected by Mulla’s formula [41] considering mosquito population in the untreated control plot before and after treatment.

Our field trial findings show that crude carvacrol-rich oregano EO works efficiently under field conditions against mosquito larvae either alone or as a mixture with both authorized insecticides, a biopesticide (*B.t.i.*) and an insect growth regulator (diflubenzuron). Because EOs contain a mixture of active components capable of acting on insects at different target sites or with different modes of action, they are less prone to leading to resistance compared to synthetic insecticides [24,28,42,43], which represents a potential advantage of using oregano EO in mixtures with synthetic insecticides. The combined application of oregano EO with *B.t.i.* may be useful in cases where the activity of *B.t.i.* is affected by biological or environmental factors, such as reduced feeding activity of last instar larvae [44] and/or reduced availability of toxins to mosquitoes in polluted waters due to increased toxin adsorption by suspended organic matter particles [45,46]. From the perspective of developing novel products for mosquito larval control with multiple mechanisms of action to improve overall bioactivity and avoid resistance induction, it may be noted that apart from the direct toxic effect on larvae, the application of oregano EO in the breeding sites may exert a repellent activity on the egg-laying females, as has been determined in the field trial of Evergetis et al. [40]. On the other hand, the concomitant use of conventional insecticides with EO may mitigate possible drawbacks of the single oil application, such as the relatively high cost of production, significant variation in biological action due to variable composition, and quick evaporation in aqueous environments, which may reduce the residual activity [29,47,48].

Overall, the study herein provides experimental evidence of the capacity of carvacrol-rich oregano EO for efficient application as a mixture with diflubenzuron or *B.t.i.* in road drains against *Ae. albopictus* and *Cx. pipiens*, resulting in immature population reduction for a 2–3 week period. Although the ability of botanicals to synergize with synthetic insecticides and biological control agents against mosquito larvae has been studied on the laboratory scale, field studies investigating their performance under actual conditions are particularly scarce. Considering the relatively higher cost of botanical insecticides and the limitations involved with their actual application, this integrated use of botanicals with other synthetic or bio- insecticides could prove an economically viable option for both the pesticide industry and users.

## Figures and Tables

**Figure 1 insects-13-00977-f001:**
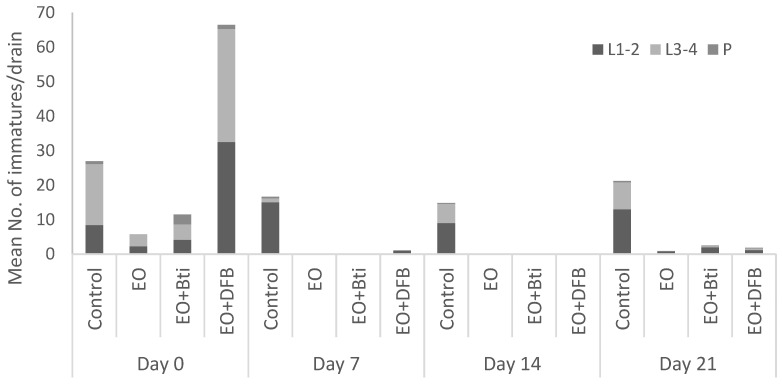
Immature mosquito population (1st–2nd and 3rd–4th instar larvae and pupae) of *Cx. pipiens* pre-treatment (Day 0) and post-treatment (Days 7, 14, and 21) with oregano essential oil (EO), oregano essential oil + diflubenzuron (EO+DFB), and oregano essential oil + *B.t.i.* (EO+*B.t.i.*) in road drains in Crevalcore, Italy, during July 2019.

**Figure 2 insects-13-00977-f002:**
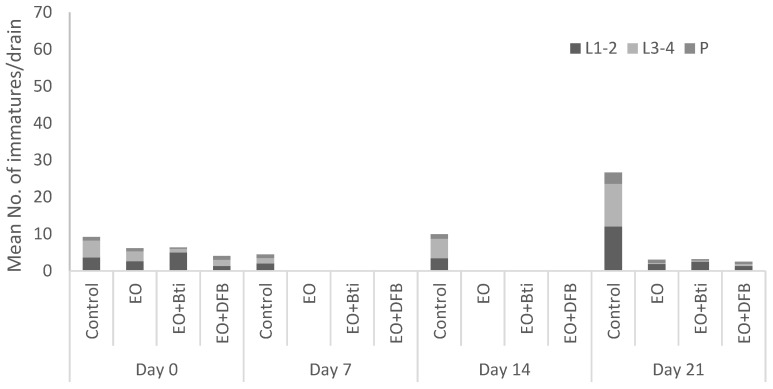
Immature mosquito population (1st–2nd and 3rd–4th instar larvae and pupae) of *Ae. albopictus* pre-treatment (Day 0) and post-treatment (Days 7, 14, and 21) with oregano essential oil (EO), oregano essential oil + diflubenzuron (EO+DFB), and oregano essential oil + *B.t.i.* (EO+*B.t.i.*) in road drains in Crevalcore, Italy, during July 2019.

**Figure 3 insects-13-00977-f003:**
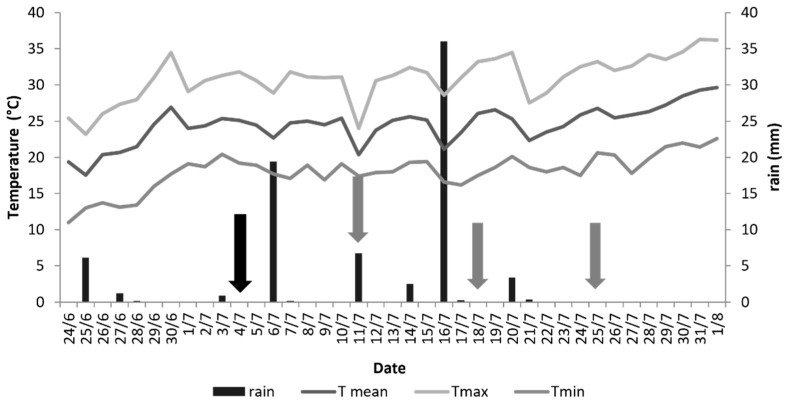
Meteorological data (min, max, and mean Temperature (°C) and rainfall (mm)) during the field study (4–25 July 2019) in Crevalcore, Italy. Data were obtained from the closest weather station, located 1.7 Km from the study area (Sant’ Agata Bolognese weather station)**.** The black arrow indicates Day 0 (treatment day); grey arrows indicate Days 7, 14, and 21 post-treatment.

**Table 1 insects-13-00977-t001:** Mean % of immature population reduction (corrected using Mulla’s formula) (+/− SEM) of *Cx. pipiens* and *Ae. albopictus* at 7, 14, and 21 days post-treatment with oregano essential oil (EO), oregano essential oil + diflubenzuron (EO+DFB), and oregano essential oil + *B.t.i.* (EO+*B.t.i.*) in road drains in Crevalcore, Italy, during July 2019.

Time	Product		*Cx. pipiens*			*Ae. albopictus*			Total		
		N	Mean%	SEM	*p* Value	Mean %	SEM	*p* Value	Mean %	SEM	*p* Value
Day 7	EO	7	100.0	0.0	0.102	100.0	0.0	1.000	100.0	0.0	0.102
	EO+*B.t.i.*	6	100.0	0.0		100.0	0.0		100.0	0.0	
	EO+DFB	6	97.6	2.0		100.0	0.0		97.6	2.0	
Day 14	EO	7	100.0	0.0	0.338	100.0	0.0	1.000	100.0	0.0	0.338
	EO+*B.t.i.*	6	97.4	2.6		100.0	0.0		98.6	1.4	
	EO+DFB	6	100.0	0.0		100.0	0.0		100.0	0.0	
Day 21	EO	7	81.0	15.6	0.933	83.1	5.8	0.806	75.4	9.4	0.143
	EO+*B.t.i.*	6	72.4	18.8		82.7	9.5		76.0	11.6	
	EO+DFB	6	96.5	2.2		78.4	13.2		95.4	2.8	

For *p* > 0.05, no significant differences in population reduction rate between applications were identified for each sampling day post treatment using the non-parametric Kruskal–Wallis test (K samples).

## Data Availability

All relevant data are within the paper.

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
