# Peer review of "Efficacy Evaluation of Oregano Essential Oil Mixed with Bacillus thuringiensis israelensis and Diflubenzuron against Culex pipiens and Aedes albopictus in Road Drains of Italy"

_insects, 2022, doi:10.3390/insects13110977_

Round 1

Reviewer 1 Report

Materials & Methods:  

please provide more details with the dimensions of the road drains and the used field doses of diflubenzuron and Bti compared to the ones used to the present study.

the use of a set of road drains treated only with diflubenzuron/Bti (positive control) could provide with more strong evidence (make comparisons easier)

Author Response

Reviewer 1:

Materials & Methods: 

please provide more details with the dimensions of the road drains and the used field doses of diflubenzuron and Bti compared to the ones used to the present study.

Response: The dimensions of the road drains were 40 x 40 cm; this is clarified in the revised manuscript (line 147). In lines 150-151 it is also clarified that the road drains were closed hydraulic systems maintaining highly polluted water with suspended and floating materials.

Regarding the applied doses, diflubenzuron-based formulation was applied at the recommended dose on the label for closed hydraulic systems. The applied dose of B.t.i. formulation was 10 times higher than the average dose recommended on the label for open polluted water bodies due to highly polluted water in closed hydraulic systems. The high dose of B.t.i. was chosen because we know that B.t.i. in drains is very weak, and the max dose in the label is referred to open polluted water body, not to closed drains. Hence, the following text is added in the revised MS (lines 155-158): “Diflubenzuron-based formulation was applied at the recommended dose on the label for closed hydraulic systems. The applied dose of B.t.i.formulation was 10 times higher than the average dose recommended on the label for open polluted water bodies due to highly polluted water in closed hydraulic systems”.

the use of a set of road drains treated only with diflubenzuron/Bti (positive control) could provide with more strong evidence (make comparisons easier)

Response:

DFB and B.t.i were not tested individually because their performance in road drains against mosquitoes was well known. Moreover, this study aimed to assess the effectiveness of carvacrol rich crude oregano EO as a mixture with diflubenzuron and B.t.i. in road drains against Ae. albopictus and Cx. pipiens populations, focusing on the impact of DFB and B.t.i. on the performance of oregano EO. This is addressed in the scope of the revised MS (lines 129-130): “…; the study focused on the impact of diflubezuron and B.t.i. on oregano EO performance”.

Reviewer 2 Report

This study examines the use of an essential oil with an IGR or microbial insecticide. The measure used for assessment is population reduction in drainage infrastructure. The study period is three weeks.

The introduction is well written, provides plenty of background on the ecology of the species in the area of study, a good explanation of the current state of larvicidal resistance and the normal modes of treatment.

The methods need some additional information.

What was your threshold for a drain to be considered "colonized? How many mosquitoes needed to be present when initially dipped?

Is the experimental design appropriate? Please justify your design in the methods section. I have two concerns that I would like to see addressed. First, if testing combinations, isn't it necessary to have each component tested individually? The EO is tested individually but the DFB and Bti are not. Why not? Second, the time period of three weeks seems very short. Notably in your data, the DFB at 21 days starts to look superior but the experimental period ends while mortality is still generally high. As it is suggested later in the manuscript that a benefit of a combination is that they can reduce costs, isn't the biggest cost savings the reduction of labor that would be seen by a lengthy retreatment period?

The data analysis seems appropriate but could also use additional detail to make the analysis more clear. Was Mulla's formula applied to averaged data for each treatment type or applied to individual location counts? Lines 165-167 indicates means, line 170 says "in total, per drain" so it is a little unclear. However, if the calculations are per drain, then it will be necessary to specify which of the 10 control locations was used for Mulla's.

The results are well presented but ultimately show no significant differences between the treatments over the time period studied. This leaves the Authors in a tough position where a good study shows no significant result. This happens and is quite acceptable; it is science. However, is such situation, it is critical not to overreach with the conclusion or discussion. It is impossible to conclude from this whether there was any benefit to the 2 combinations so I would suggest the Authors limit speculation on the utility of this treatment. Certainly there is nothing that can be learned from this study regarding the impact on resistance management which was one of the issues raised in the introduction. Also, in Lines 283-284, it is stated "results herein verifying the high synergistic activity of phytochemicals in combination with currently used synthetic insecticides or bio-insecticides…” This line is not correct and needs revision. This study does not show synergistic activity. The KW test shows no difference between EO, EO&DFB and EO&BTI so it is inappropriate to claim “results herein verify.” While this study does not support this statement, the Authors are correct that the EO literature does.

Author Response

Reviewer 2:

This study examines the use of an essential oil with an IGR or microbial insecticide. The measure used for assessment is population reduction in drainage infrastructure. The study period is three weeks.

The introduction is well written, provides plenty of background on the ecology of the species in the area of study, a good explanation of the current state of larvicidal resistance and the normal modes of treatment.

The methods need some additional information.

What was your threshold for a drain to be considered "colonized? How many mosquitoes needed to be present when initially dipped?

Response: The presence of at least one mature larva or pupa per water sample dipper was sufficient to consider the drain colonized by mosquito larvae/pupae. This is clarified in lines 148-149 of the revised manuscript.

Is the experimental design appropriate? Please justify your design in the methods section. I have two concerns that I would like to see addressed. First, if testing combinations, isn't it necessary to have each component tested individually? The EO is tested individually but the DFB and Bti are not. Why not?

Response: DFB and B.t.i were not tested individually because their performance in road drains against mosquitoes was well known. Moreover, this study aimed to assess the effectiveness of carvacrol rich crude oregano EO as a mixture with diflubenzuron and B.t.i. in road drains against Ae. albopictus and Cx. pipiens populations, focusing on the impact of DFB and B.t.i. on the performance of oregano EO. This is addressed in the scope of the revised MS (lines 129-130): “…; the study focused on the impact of diflubezuron and B.t.i. on oregano EO performance”.

Second, the time period of three weeks seems very short. Notably in your data, the DFB at 21 days starts to look superior but the experimental period ends while mortality is still generally high. As it is suggested later in the manuscript that a benefit of a combination is that they can reduce costs, isn't the biggest cost savings the reduction of labor that would be seen by a lengthy retreatment period?

Response: Indeed, according to our results it would be worth studying the persistent action particularly of EO+DFB for a longer period. Nevertheless, this study aimed to provide for the first time, baseline data from the field on the capacity of oregano EO to be applied efficiently as mixture with DFB and Bti against mosquitoes. The 3-week interval in the study set up was chosen as a follow up to our previous results (Evergetis et al. 2018) one year earlier at the same area when the same oregano oil at the same dose was proved as efficient larvicidal agent against Cx. pipiens and Ae. albopictus for 2 weeks (lines 125-126). Moreover, in practice the persistent action of DFB and Bti in drains usually does not exceed 3 weeks and in case of operational conditions larval mortality should be sufficiently high during the interval between two treatments, hence the 3-week period seems suitable. Finally, the economic benefit for the industry and the users of mixing relatively costly botanicals with conventional insecticides, is already mentioned in lines 278, 287-290.

The data analysis seems appropriate but could also use additional detail to make the analysis more clear. Was Mulla's formula applied to averaged data for each treatment type or applied to individual location counts? Lines 165-167 indicates means, line 170 says "in total, per drain" so it is a little unclear. However, if the calculations are per drain, then it will be necessary to specify which of the 10 control locations was used for Mulla's.

Response: We clarify that for the % of population reduction for each treatment, we used Mulla’s formula (correction with control) considering the number of mosquitoes in each drain pre- (T1) and post- (T2) treatment, and the mean number of mosquitoes per drain in the control site pre- (C1) and post- (C2) treatment.

This is clarified in the text of the revised MS (lines 177-180) as follows: “C1 is the mean number of mosquitoes per drain at the control site pre-treatment; C2 is the mean number of mosquitoes per drain at the control site post-treatment; T1 is the number of mosquitoes in each drain at the treatment site pre-treatment, and T2 is the number of mosquitoes in each drain at the treatment site post-treatment”.

Then, as already mentioned (lines 181-185), the mean (corrected) % of population reduction in each species and in total (both species) per drain, was calculated and the data were analyzed using the non-parametric K-W test to determine significant differences between treatments.

The results are well presented but ultimately show no significant differences between the treatments over the time period studied. This leaves the Authors in a tough position where a good study shows no significant result. This happens and is quite acceptable; it is science. However, is such situation, it is critical not to overreach with the conclusion or discussion. It is impossible to conclude from this whether there was any benefit to the 2 combinations so I would suggest the Authors limit speculation on the utility of this treatment. Certainly there is nothing that can be learned from this study regarding the impact on resistance management which was one of the issues raised in the introduction. Also, in Lines 283-284, it is stated "results herein verifying the high synergistic activity of phytochemicals in combination with currently used synthetic insecticides or bio-insecticides…” This line is not correct and needs revision. This study does not show synergistic activity. The KW test shows no difference between EO, EO&DFB and EO&BTI so it is inappropriate to claim “results herein verify.” While this study does not support this statement, the Authors are correct that the EO literature does.

Response: Following your suggestions, we removed most of the text in the discussion regarding resistance leaving only lines 264-267 where the potential advantage of using oregano EO in mixtures with insecticides is mentioned. We also removed the phrase “Results herein verifying…”. References are revised accordingly.